# Suppression of Cross-Coupling Effect of Hybrid Permanent Magnet Synchronous Motor with Parallel Magnetic Circuit

**Xiao He and Guangqing Bao ***

College of Electrical and Information Engineering, Lanzhou University of Technology, Lanzhou 730050, China; hxelectricity@163.com
* Correspondence: Baogq03@163.com

**Abstract:** Hybrid permanent magnet synchronous motor (HPMSM) has attracted increased attention in recent years due to its adjustable air gap flux. However, as a result of the cross-coupling effect of high- and low-coercive permanent magnets, the precise magnetic adjustment of HPMSM has become increasingly difficult. In order to weaken the cross-coupling effect, two methods of adding magnetic barrier and exciting coil are adopted in this paper. First, the equivalent magnetic circuit model is established, and the theoretical rationality of the weakening method is analyzed. Second, the electromagnetic performance of two weakening methods are analyzed based on the finite element analysis. Finally, the rationality of the theoretical analysis is verified, which provides the structure basis for the precise magnetic adjustment of the hybrid permanent magnet motor.

**Keywords:** hybrid permanent magnet motor; cross-coupling effect; precise magnetic adjustment; finite element analysis





## 1. Introduction

The permanent magnet synchronous motor (PMSM) is widely used in aerospace and electric vehicle drive systems due to its simple structure, reliable operation, and high-power density [1]. However, due to the use of high-coercive permanent magnets in the motor, the air gap flux is difficult to adjust, which makes the speed range of the motor small. When the motor is required to operate in a wide speed range, a continuous direct-axis weak magnetic current is required. This method has some drawbacks: (1) Extra copper consumption is produced, and (2) large weak magnetic current may cause irreversible demagnetization of the permanent magnet.

German scholar V. Ostovic first proposed the concept of "memory motor" in 2001, in which online pole writing is introduced into an ordinary permanent magnet synchronous motor [2]. The low-coercive permanent magnet is used rather than the high-coercive permanent magnet. In addition, the low-coercive permanent magnet is magnetized and demagnetized to achieve the purpose of regulating the air gap flux. Subsequently, many scholars have conducted a large amount of research on memory motor, and the topology of memory motor has been continuously innovated [3–5]. However, due to the low magnetic energy product of low-coercive permanent magnets, the power density of the motor is limited and the power index is poor.

In recent years, hybrid permanent magnets have been proposed to solve the low power density problem of memory motors. In 2010, Japanese scholar Kazuto Sakai proposed an "U" hybrid permanent magnet memory motor [6], analyzed the operation principle of the motor, and verified the rationality of the prototype. Dr. Yang Hui of Southeast University proposed the combination of split-tooth cursor motor with the memory motor [7], to achieve the low-speed and high-torque output of the motor through magnetic field modulation. An "Y" type hybrid permanent magnet memory motor is proposed by Professor Chen Yangsheng of Zhejiang University. In addition, using the finite element method, the magnetic characteristics and design methodology of the proposed motor are studied and

analyzed [8]. Then, to solve the problem of the forward magnetization difficulty of the motor, Dr. Zhou Yubin et al. proposed an optimized topology to increase the magnetic barrier in the rotor [9]. The group of Zhu ziqiang professors from the University of Sheffield, UK, made a comparative study of the series-parallel hybrid permanent magnet memory motor [10]. The magnetic regulating principle of the two structures is analyzed, and the performance of the motor is tested. In the above motor structure, due to the cross-coupling effect between the high- and low-coercive permanent magnets and the unintentional demagnetization of armature current to low-coercive permanent magnets, the difficulty and accuracy of low-coercive online magnetization and the performance of the motor will be affected. Reference [11] analyzed the cross-coupling effect between the two permanent magnets in detail by the equivalent magnetic circuit method and finite element simulation.

In this paper, the parallel magnetic circuit hybrid permanent magnet is used on the basis of the motor structure [10]. The cross-coupling effect is weakened by setting the magnetic barrier between the two permanent magnets and adding exciting coil to the low-coercive permanent magnet. The equivalent magnetic circuit model of the motor structure is established, the coupling reason is analyzed, and the improved model is simulated using the finite element software to verify the correctness of the method, which provides a theoretical basis for reducing the difficulty of magnetic regulation of hybrid permanent magnet memory motor.

## 2. Motor Topology

The motor shown in Figure 1 has eight poles and 48 slots, and the high- and low-coercive permanent magnets are V-shaped. The high-coercive permanent magnet is NdFeB, which is represented by CPM, and the low-coercive permanent magnet is AlNiCo and represented by VPM. The permanent magnet is placed in the rotor and a magnetic barrier is set to ensure the correct direction of the magnetic flux. The arrows in the figure indicate the magnetization direction of the permanent magnet. The detailed motor parameters are shown in Table 1.

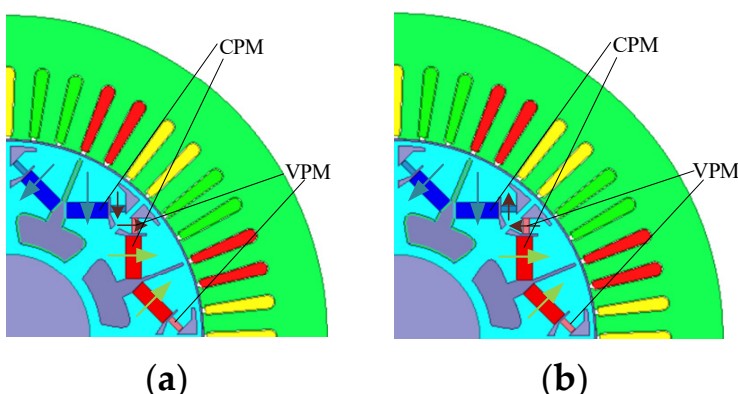

**Figure 1.** Motor topology. (**a**) Forward parallel structure; (**b**) reverse parallel structure.

**Table 1.** Motor parameters.

| Parameters | Values | Parameters | Values |
|---|---|---|---|
| Phases number | 3 | CPM thickness (mm) | 6.5 |
| Stator slots/rotor poles | 48/8 | CPM width (mm) | 25 |
| Axial length (mm) | 50.8 | VPM thickness (mm) | 7 |
| Stator outer diameter | 264 | VPM width (mm) | 25 |
| Stator inner diameter | 161.9 | Number of turns per coil | 11 |
| Rotor outer diameter | 160.44 | Air-gap length (mm) | 0.73 |
| Rotor inner diameter | 68 | Rated speed (rpm) | 1500 |

### 3. Analysis of Equivalent Magnetic Circuit and Magnetic Modulation

*3.1. Equivalent Magnetic Circuit Model*

In order to better discuss the operation mechanism of the hybrid permanent magnet with the parallel magnetic circuit, its equivalent magnetic circuit model is first analyzed.

Figure 2 shows the two equivalent magnetic circuits in the forward and reverse parallel connection of the magnetic circuit, in which, $F_C$ and $F_V$ represent the magnetic momentum of high- and low-coercive permanent magnets; $R_C$ and $R_V$ represent the magnetic resistance of high and low coercive permanent magnet coupling circuits; and their size is related to the length of the magnetic circuit when the two permanent magnets are coupled. $R_C'$ and $R_V'$ represent the residual magnetic resistance of the magnetic flux of the permanent magnet, which forms the rotor resistance with the corresponding coupling magnetic resistance; $R_g$ is the resistance of the air gap; $\Phi_C$, $\Phi_V$, and $\Phi_g$ represent the magnetic flux provided by the high- and low-coercive permanent magnets and the magnetic flux at the air gap. When the motor is not loaded, the air gap magnetic field is only established by the permanent magnet. For the forward parallel magnetic circuit (Figure 2a), these are the following relationships:

$$\Phi_g = \Phi_C + \Phi_V \tag{1}$$

$$R_s = R_g + R_C' \parallel R_V' \tag{2}$$

$$\Phi_C = \frac{F_C}{R_C + R_s \parallel R_V} - \frac{F_V}{R_V + R_s \parallel R_C} \times \frac{R_s}{R_s + R_C} \tag{3}$$

$$\Phi_V = \frac{F_V}{R_V + R_s \parallel R_C} - \frac{F_C}{R_C + R_s \parallel R_V} \times \frac{R_s}{R_s + R_V} \tag{4}$$

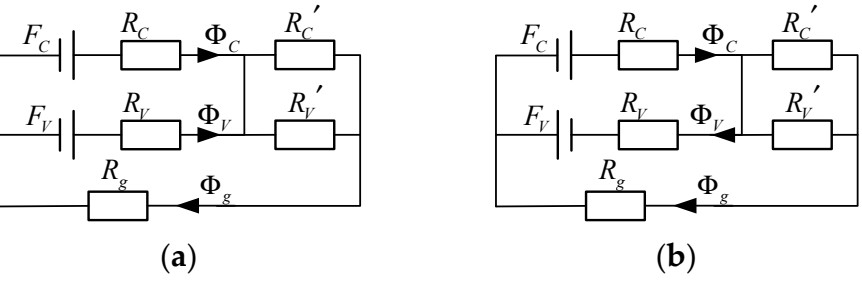

**Figure 2.** Equivalent magnetic circuits. (**a**) Forward parallel equivalent magnetic circuit; (**b**) reverse parallel equivalent magnetic circuit.

In Equation (4), it is easy to know that if the difference of magnet potential $F_C$ and $F_V$ between the high- and low-coercive permanent magnets is too large, the magnetic flux $\Phi_V$ may be negative. In this case, the high-coercive permanent magnets change the magnetization direction of the low-coercive permanent magnets due to the cross-coupling effect, which may lead to the loss of magnetization of low-coercive permanent magnets. In addition, its demagnetization reduces the effective magnetic flux, weakens the air gap magnetic density, and reduces the back-EMF during the no-load operation.

Similarly, for the reverse parallel equivalent magnetic circuit in Figure 2b, it is known that:

$$\Phi_g = \Phi_C - \Phi_V \tag{5}$$

$$R_s = R_g + R_C' \parallel R_V' \tag{6}$$

$$\Phi_C = \frac{F_C}{R_C + R_s \parallel R_V} + \frac{F_V}{R_V + R_s \parallel R_C} \times \frac{R_s}{R_s + R_C} \tag{7}$$

$$\Phi_V = \frac{F_V}{R_V + R_s \parallel R_C} + \frac{F_C}{R_C + R_s \parallel R_V} \times \frac{R_s}{R_s + R_V} \tag{8}$$

In Equation (8), the influence of the cross-coupling effect on the low-coercive permanent magnets causes the magnetic flux to increase, which is equivalent to magnetizing,

resulting in a greater current in the armature current when weakening the magnets with a low-coercive force.

### 3.2. Principle of Parallel Magnetic Circuit Magnetic Adjustment

As shown in Figure 3, the schematic diagram of the parallel permanent magnetic circuit regulation principle is given in the process of magnetizing the low-coercive permanent magnet by adjusting the magnetized pulse. It is assumed that the high- and low-coercive permanent magnets are stable and work at A and a, respectively. Therefore, the gap flux can be expressed as:

$$\Phi_g = B_A \cdot S_C + B_a \cdot S_V \tag{9}$$

where $B_A$ and $B_a$ represent the flux density of high- and low-coercive permanent magnets, respectively; $S_C$ and $S_V$ represent the cross-sectional area of high- and low-coercive permanent magnets, respectively.

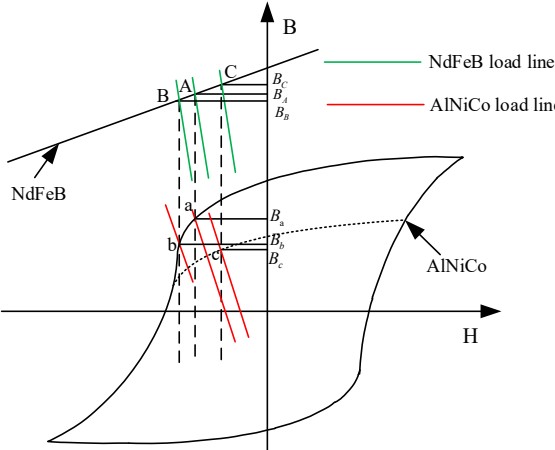

**Figure 3.** Illustration of flux-regulation mechanism of the parallel hybrid permanent magnet magnetic circuit.

When a certain demagnetizing pulse is applied to a low-coercive permanent magnet, its working point transits from point A to point B along the magnetization curve, and the working point of the high-coercive permanent magnet transits from point A to point B.

When the magnetic pulse is removed, both permanent magnets reach stability along their respective recovery lines: High-coercive permanent magnet transitions to point C, and low-coercive permanent magnet goes to point C. At this point, the gap flux is:

$$\Phi_g = B_C \times S_C + B_c \times S_V \tag{10}$$

Compared with Equation (9), the gap flux in Equation (10) reduces the difference between the recovery flux of high- and low-coercive permanent magnets, thus achieving the goal of an adjustable magnetic flux. If a larger demagnetizing pulse is applied, the gap flux decreases further.

## 4. Methods to Reduce the Cross-Coupling Effect

Due to the cross-coupling effect between the high- and low-coercive permanent magnets, which is related to the magnetic resistance of the permanent magnet circuit, if the coupling effect is too large and the high-coercive permanent magnet demagnetizes the low-coercive permanent magnet to the non-linear segment, the armature current will no longer be linear to the low-coercive permanent magnet. This makes it difficult to adjust the magnetic field, and at the same time, there will be an inaccurate and effective magnetic flux reduction, the problem of extra wastage and inefficiency. Therefore, this paper presents two solutions to this problem.

### 4.1. Lengthening the Magnetic Barrier between Two Kinds of Permanent Magnets

As shown in Figure 4, based on the original model, this paper increases the length of the magnetic barrier, increases the coupling circuit between the high- and low-coercive permanent magnets, and increases the magnetic resistance, which decreases the two magnetic resistances $R_V{}'$ and $R_C{}'$, thereby weakening the influence between the two permanent magnets. The magnetic barrier length a is the length beyond the high coercivity permanent magnet.

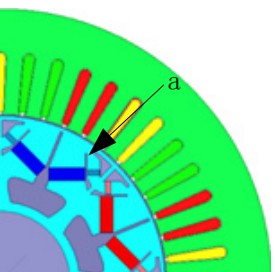

**Figure 4.** Lengthened magnetic barrier.

### 4.2. Setting the Excitation Coil Close to the Low-Coercive Permanent Magnet

Due to the existence of high-coercive permanent magnets, the working point of low-coercive permanent magnets will be greatly affected. As shown in Figure 5, an excitation winding near a low-coercive permanent magnet is proposed to counteract or weaken the coupling effect by magnetizing the low-coercive permanent magnet in both the positive and negative directions.

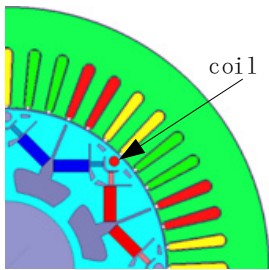

**Figure 5.** Add excitation winding.

The equivalent magnetic circuit is shown in Figure 6. $F_e$ is the equivalent magnetic motive force generated by the excitation current. The magnetic motive force interacts with the magnetic coercive force $F_C$ and $F_V$ of the high- and low-coercive force in the magnetic circuit. To a certain extent, it can magnetize and demagnetize the low-coercive force permanent magnet to weaken the coupling effect. In the model of the positive parallel magnetic circuit shown in Figure 6a, the magnetic flux $\Phi_{ev}$ generated by the excitation potential in the magnetic circuit of a low-coercive permanent magnet is known as:

$$\Phi_{ev} = F_e \cdot \frac{1}{(R_C{}' \parallel R_V{}') \parallel (R_r + R_g) + R_e} \times \frac{R_r + R_g}{(R_C{}' \parallel R_V{}') + R_r + R_g} \times \frac{R_C{}'}{R_C{}' + R_V{}'} \quad (11)$$

in which, $R_C{}'$ and $R_V{}'$ represent the magnetic resistance of high- and low-coercive permanent magnets in the coupling effect; $R_r$ is the magnetic resistance parallel value of the uncoupled circuit; and $R_e$ is the excitation resistance. Similarly, in Figure 6b, the flux $\Phi_{ev}{}'$ generated by the excitation magnetomotive force is equal to $\Phi_{ev}$. It is known from Equation (11) that the excitation potential $F_e$ can indeed change the magnetic flux of a low-coercive permanent magnet, change its operating point, and weaken the cross-coupling effect in both the forward and reverse parallel models.

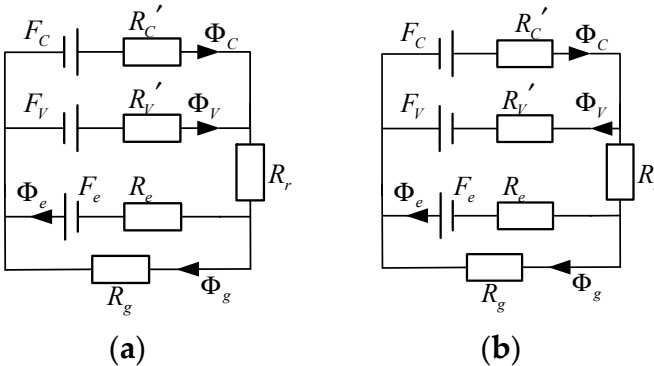

**Figure 6.** Equivalent magnetic circuit with excitation magnetomotive force. (**a**) Forward parallel equivalent magnetic circuit; (**b**) reverse parallel equivalent magnetic circuit.

## 5. Electromagnetic Analysis and Verification

Based on the above theoretical analysis, the two-dimensional electromagnetic simulation of each model is carried out using the finite element simulation software and freezing permeability method.

### 5.1. Lengthened Magnetic Barrier

Increasing the magnetic barrier length indicates an increase in the magnetic circuit between the two permanent magnets, increase in the coupling reluctance, and reduction of the magnetic flux, in order to weaken the cross-coupling effect.

#### 5.1.1. No-Load Analysis

Figure 7 shows the magnetic line distribution of different length magnetic barriers in the forward parallel and no-load state of the permanent magnets of the motor. It can be seen from the diagram that the magnetic flux of the motor increases with the length of the magnetic barrier, which indicates that the coupling effect between the two permanent magnets is weakened and more air gap flux is produced due to the increase of the coupling magnetic circuit. When A = 13 mm, some magnetic lines pass through the magnetic barrier in Figure 7d, since the magnetic barrier is longer and covers part of the high-coercive permanent magnet magnetic circuit.

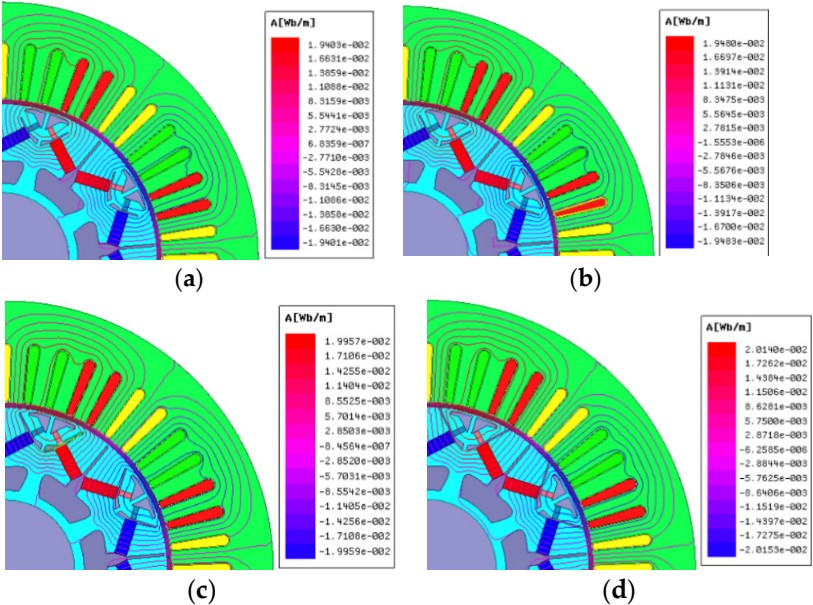

**Figure 7.** Distribution of magnetic lines with different lengths of magnetic barriers under the no load operation. (**a**) a = 5 mm; (**b**) a = 7 mm; (**c**) a = 10 mm; (**d**) a = 13 mm.

The magnitude of back EMF is directly related to the flux density, and Figure 8 shows the effect of the length of magnetic barrier on EMF. It is easy to see that the larger the a, the larger the amplitude of the back EMF wave and the larger proportion of low-order harmonics. It is due to the decrease of cross-coupling effect and the increase of effective magnetic flux that the amplitude of the back EMF is larger and the sinusoidal degree is higher.

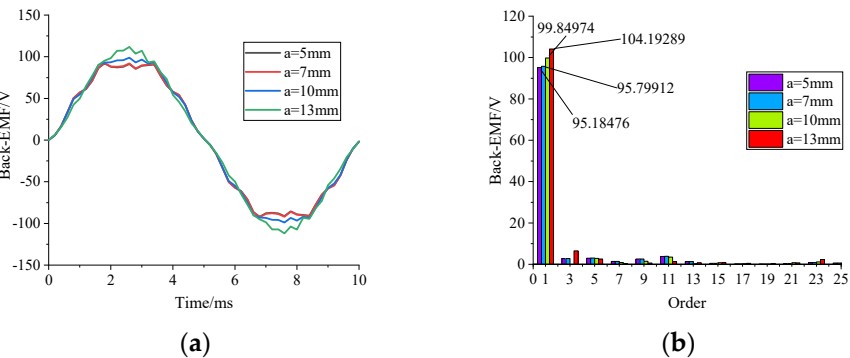

**(a)**　　　　　　　　　　　　　　　　　　**(b)**

**Figure 8.** Back EMF with different lengths of magnetic barriers. (**a**) Back EMF waveform; (**b**) harmonic spectra.

Figure 9 shows the magnetic density perpendicular to the direction of magnetization on a low-coercive permanent magnet. From Figure 9, it can be seen that when the magnetic barrier increases, the magnetic density of the low-coercive permanent magnet increases, the working point moves up, and the cross-coupling effect decreases.

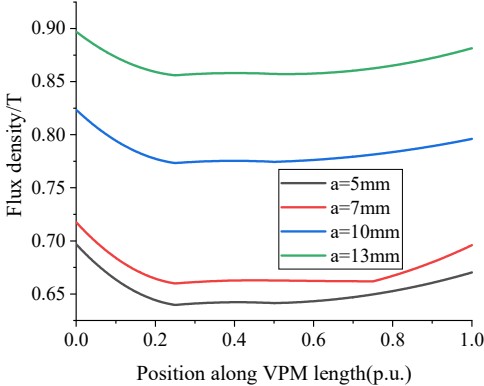

**Figure 9.** Low-coercive permanent magnetic flux density.

Figure 10 shows the air gap flux density waveforms and spectrum analysis for four different barrier lengths. Moreover, it shows that increasing the coupling magnetic circuit between the two permanent magnets can effectively reduce the cross-coupling effect. Figure 11 shows the distribution of the magnetic lines of the motor when two permanent magnets are inversely parallel. It is easy to see that as the length of the magnetic barrier increases, the flux density of the motor decreases. When the magnetic barrier is very short, the low-coercive permanent magnet increases the working point due to the coupling effect, and produces more reverse magnetic flux. As a result, the additional magnetic flux of the high-coercive permanent magnet forms a loop with the low-coercive permanent magnet, which increases the magnetic flux of the rotor and further reduces the air gap flux density. As the magnetic barrier increases, the coupling effect decreases, the working point of the low-coercive permanent magnet moves down, and the flux decreases, the loop flux decreases, the rotor flux decreases, thus the air gap flux increases. The reason why the magnetic line passes through the magnetic barrier in Figure 11d is similar to the previous one. Figure 12 shows the air gap flux density and its spectrum analysis of the motor. With

the increase of magnetic barrier, the air gap flux density increases gradually, which verifies the previous analysis.

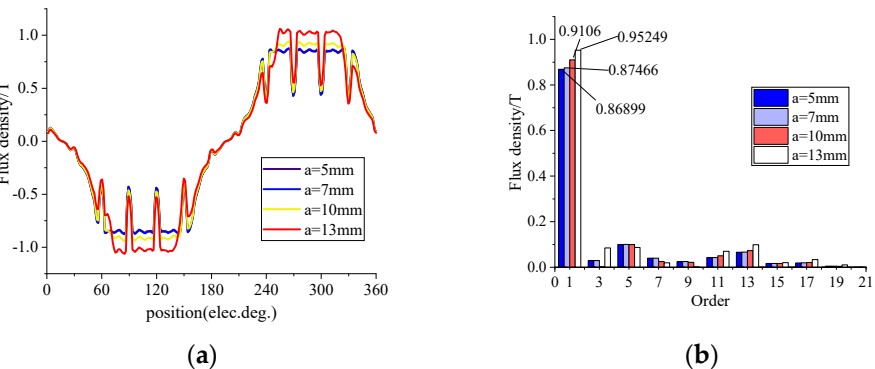

**Figure 10.** Air gap flux density with different barrier lengths. (**a**) Waveform of air gap flux density; (**b**) spectrum analysis.

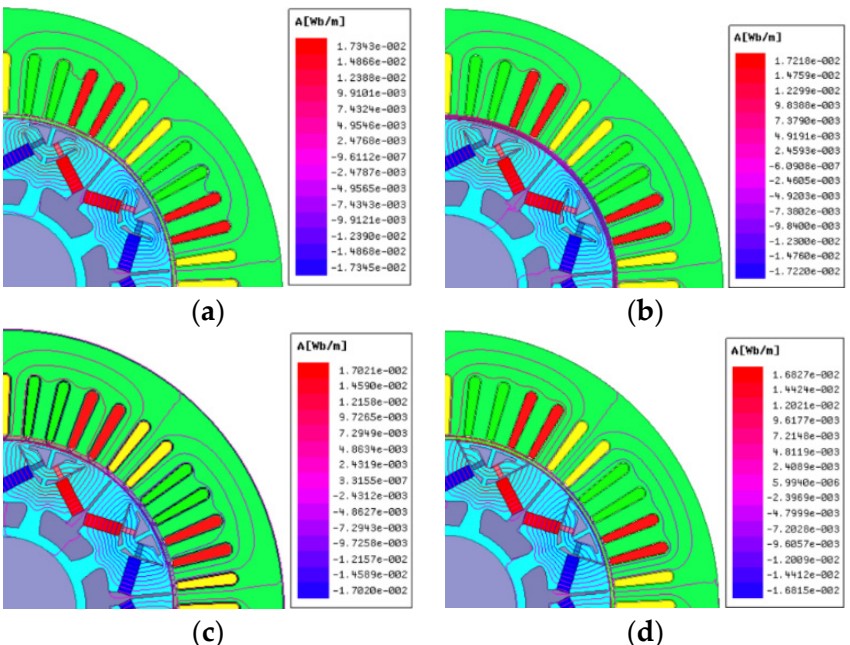

**Figure 11.** Distribution of magnetic lines with different lengths of magnetic barriers under the no load operation. (**a**) a = 5, (**b**) a = 7, (**c**) a = 10, (**d**) a = 13.

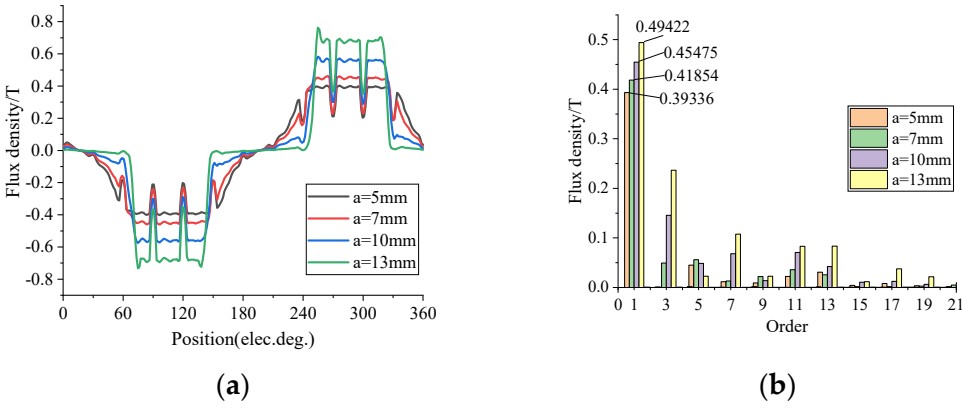

**Figure 12.** Air gap flux density with different lengths of magnetic barriers. (**a**) Waveform of air gap flux density; (**b**) harmonic spectra.

Figure 13 shows the magnetic density perpendicular to the direction of magnetization on a low-coercive permanent magnet. As the coupling effect decreases, the ability of the high-coercive permanent magnets to raise the operating point of low-coercive permanent magnets decreases, thus its flux density decreases.

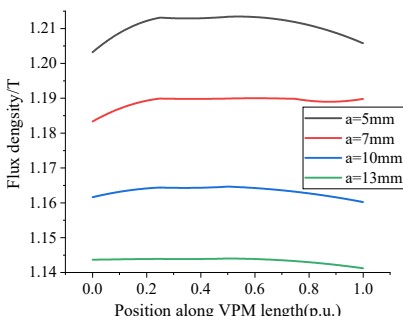

**Figure 13.** Low-coercive permanent magnet flux density.

The no-load back EMF in Figure 14 still satisfies the increase with the increase of the magnetic barrier, demonstrating that the coupling effect has been weakened again.

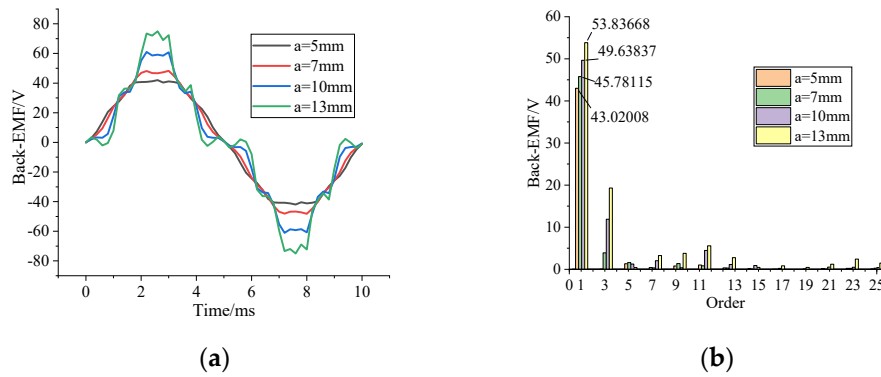

(**a**)　　　　　　　　　　　(**b**)

**Figure 14.** Back EMF with different lengths of magnetic barriers. (**a**) Waveform of back EMF; (**b**) harmonic spectra.

5.1.2. Load Analysis

Figure 15 shows the output torque of the motor when the 236A current is applied to the armature winding. It is easy to see that whether the two permanent magnets are forward parallel or reverse parallel, the output torque trend of the motor meets the above analysis, which proves that increasing the length of the magnetic barrier is effective to weaken the coupling effect.

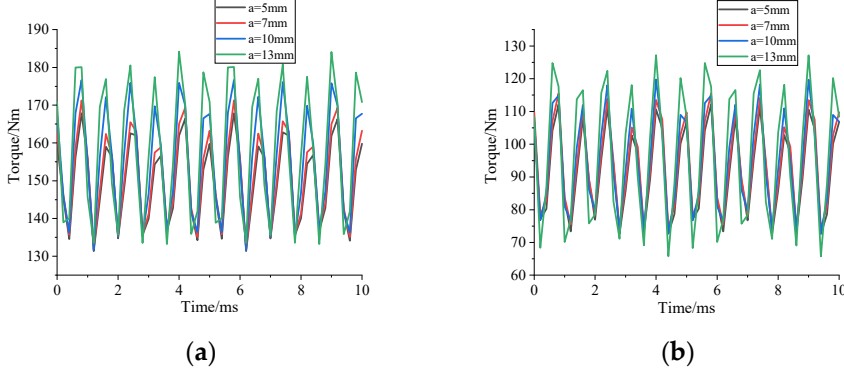

(**a**)　　　　　　　　　　　(**b**)

**Figure 15.** Motor output torque. (**a**) Output torque with the forward parallel permanent magnet; (**b**) output torque with the forward parallel permanent magnet.

### 5.2. Adding the Excitation Winding

The excitation winding is equivalent to a compensation winding of the low-coercive permanent magnet, which compensates the influence caused by the coupling-effect.

#### 5.2.1. No-Load Analysis

When two permanent magnets are in forward parallel, the VPM operating point moves down due to the cross-coupling effect. When the current is passed in the excitation winding, the coupling-effect is weakened and the VPM operating point is moved back. The waveform in Figure 16 conforms to the above analysis. However, due to the small current, the VPM magnetic flux changes are not clear. The voltage waveform shown in Figure 17 also confirms the previous inference. Figure 18 is the air gap flux density of the motor under different excitation currents. When the permanent magnets are in forward parallel, the excitation current will only affect the VPM working point, but has little effect on the air gap magnetic density. Since the air gap flux density is not affected by the excitation current, the no-load back EMF is also basically unchanged.

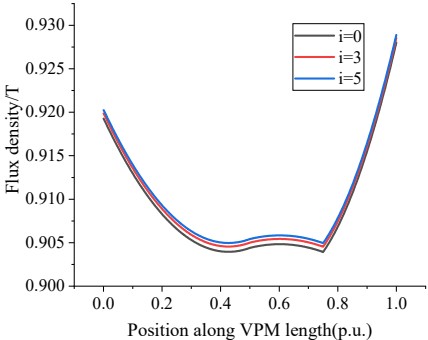

**Figure 16.** Flux density of VPM.

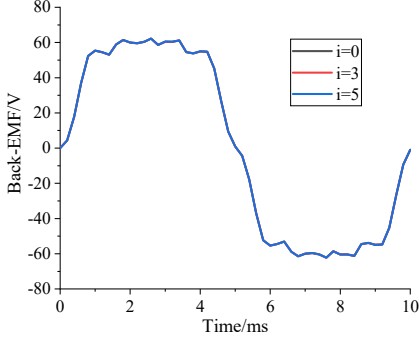

**Figure 17.** Voltage Waveform of back EMF.

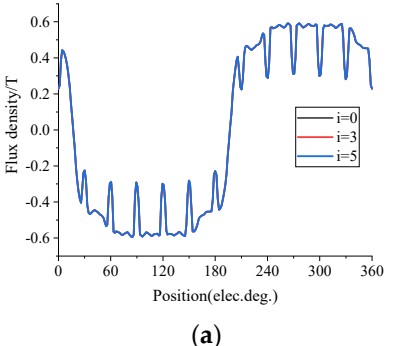

(**a**)

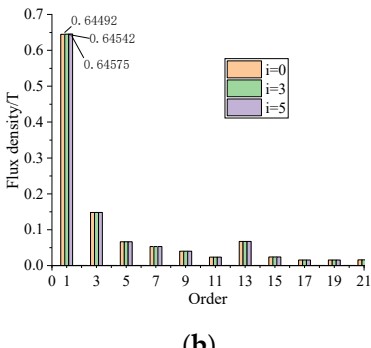

(**b**)

**Figure 18.** Air gap flux density of the motor under different excitation currents. (**a**) Waveform of air gap flux density; (**b**) harmonic spectra.

The above analysis shows that when the permanent magnet is in forward parallel, the coupling effect can be eliminated by adding an excitation winding, but it has no effect on the air gap flux density.

Figure 19 shows the working flux density of VPM under different excitation currents when the permanent magnets are reverse parallel and the motor is in the no-load operation. With the increase of the current, the working point of VPM decreases further, which indicates that the coupling effect is weakened. Figure 20 shows the no-load air gap flux density under different excitation currents. Due to the decrease of the coupling effect between the two permanent magnets, the VPM working point moves down and the magnetic flux decreases, thus the air gap flux density increases slightly. The air gap flux density of 3A and 5A in the excitation winding is almost the same, which indicates that the excitation winding has little influence on the air gap flux density. After the current is applied to the excitation winding, the air gap flux density increases, thus the no-load back EMF of the motor also increases, as shown in Figure 21.

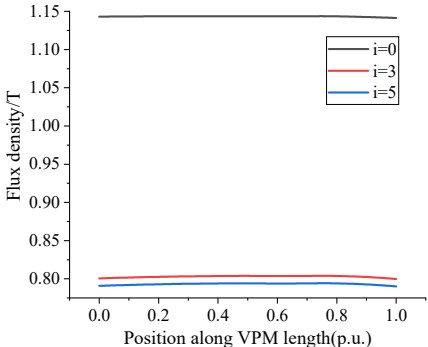

**Figure 19.** Flux density of VPM under different excitation currents when the permanent magnets are reverse parallel.

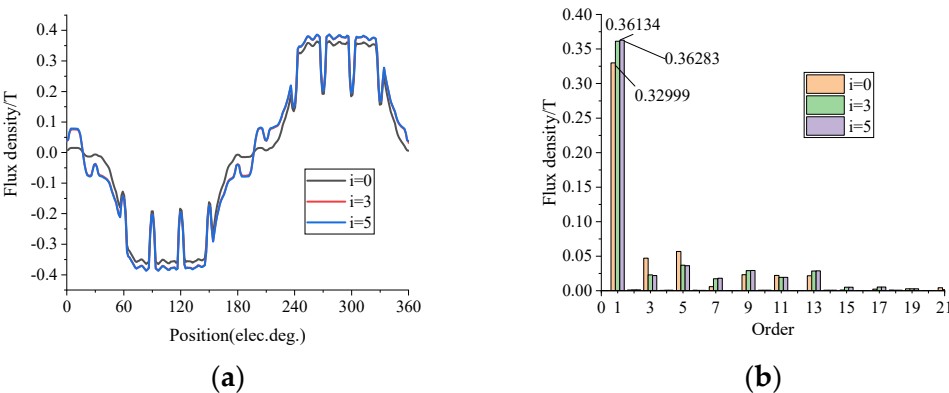

**(a)**                                           **(b)**

**Figure 20.** Air gap flux density. (**a**) Waveform of air gap flux density; (**b**) harmonic spectra.

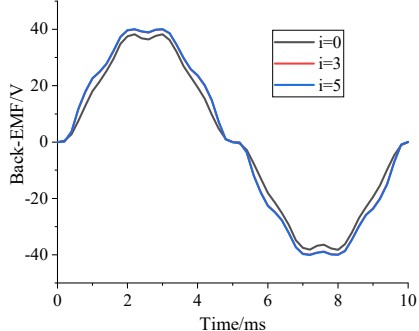

**Figure 21.** Waveform of back EMF.

### 5.2.2. Load Analysis

Figure 22 shows the motor output torque when the 236A current is applied to the motor armature. It can be seen from the above analysis that when the permanent magnets are in forward parallel, the excitation current has little effect on the air gap flux density, thus the output torque of the motor is almost the same. When the permanent magnets are in reverse parallel, the larger the excitation current is, the larger the output torque is, which is also in line with the above analysis.

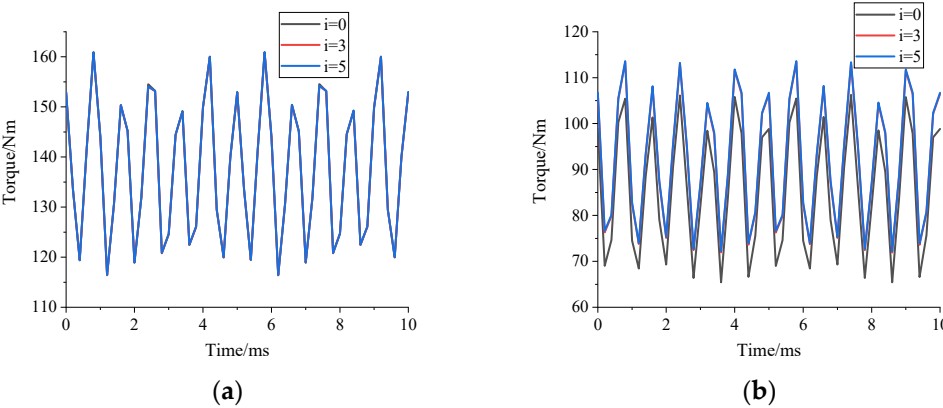

(**a**) 　　　　　　　　　　　　　　　　　　　(**b**)

**Figure 22.** Motor output torque. (**a**) Output torque with the forward parallel permanent magnet; (**b**) Output torque with the reverse parallel permanent magnet.

### 6. Conclusions and Shortcomings

The interaction between the high- and low-coercive permanent magnets in the hybrid permanent magnet memory motor, caused by the cross-coupling effect, not only leads to a large amount of flux leakage, but also affects the difficulty and accuracy of adjusting the low-coercive permanent magnets. In this paper, two methods are proposed to reduce the cross-coupling effect, which include increasing the length of magnetic barrier (increase the coupling reluctance) between the two permanent magnets and adding excitation winding to low-coercive permanent magnets. The longer the magnetic barrier, the weaker the coupling effect and the larger the air gap magnetic density, thus the larger the no-load back potential and output torque. The addition of excitation winding counteracts the effect of high-coercive permanent magnets by magnetizing and demagnetizing low-coercive permanent magnets with external circuits. The larger the current, the larger the gap magnetic density. However, attention should be paid to the design of magnetic circuits. The results of the two methods are in agreement with those of the finite element simulation, which verifies the rationality of the two methods.

Due to the fact that the above is both a theoretical and simulation analysis, it has some limited applications in practice:

(1) When setting the magnetic barrier, the original magnetic circuit and the magnetic barrier of the high-coercive permanent magnet should be avoided.
(2) Since the excitation winding is placed on the rotor, it must be equipped with a commutation device. Moreover, the additional loss and heat dissipation of the winding must be considered.
(3) The two methods inevitably lead to the complex structure of the motor, which is not easy to process.

**Author Contributions:** Conceptualization, X.H. and G.B.; Methodology, X.H. and G.B.; Software, X.H.; Validation, X.H.; Writing-Review and Editing, X.H. and G.B. All authors have read and agreed to the published version of the manuscript.

**Funding:** This research was founded jointly by the National Natural Science Foundation of China (No.51967012) and the Key Research and Development Program of Gansu Province (No. 20YF8GA055).

**Institutional Review Board Statement:** Not applicable.

**Informed Consent Statement:** Not applicable.

**Data Availability Statement:** Not applicable.

**Conflicts of Interest:** The authors declare no conflict of interest.

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
