# Peer review of "Suppression of Cross-Coupling Effect of Hybrid Permanent Magnet Synchronous Motor with Parallel Magnetic Circuit"

_wevj, doi:10.3390/wevj13010011_

Round 1

Reviewer 1 Report

This paper mainly uses simple and effective circuit models to reduce cross-coupling effects. The work of this paper is practical and logical. There are some problems to be further improved as well:

1.      The number of references is a little bit poor, and some references are not relative to your work.
2.      There are format errors in the manuscript, such as, Table1’s format ,’Figure9’ ‘Figure 10’ without a uniform standard and the lack of punctuation in formulas. Some pictures in your paper are indistinct, such as figure 7, figure 11.
3.      Some control experiments should be added to compare with other state-of-art design to show the advantages of your design.

I suggest that this paper modify with the given comments.

Author Response

1.I have modified it, removed the less relevant papers, but also added some papers.

2.The header font of Table 1 has been bold. For Figures 9 and 10, I used the uniform standard.
Figure 7 shows the distribution of the magnetic lines of the two permanent magnets in positive parallel. It can be noted that the magnetic fluxes of the two permanent magnets make up the air gap fluxes together.
Figure 11 shows the distribution of magnetic lines when two permanent magnets are inversely parallel. It is obvious that the magnetic flux of a high-coercive permanent magnet passes through a low-coercive permanent magnet. When the length of the magnetic barrier is different, the distribution of magnetic lines and the magnetic density are different.

3.Now just for principle analysis and simulation validation, no prototype has been made, and prototype will be made to validate later.

Reviewer 2 Report

Below are the technical remarks and disadvantages (recommended changes):

1. The paper conclusion is to short. Extend the conclusion to clarification of the outcomes of the research.

2. Based on which criteria is chosen the HPMSM motor topology given in paper?

3. Figure 1., 2. and 4. must be aligned with the instructions for writing the work and must be clearer.

4. Grammatical and typing errors should be corrected. For example, some formulas and abbreviations are written once in upper case letters, the other time in lower case letters. 

5. Precision explain the role of magnetic flux describe and shown in figure 2. Explain how the direction of magnetic flux changed between two types of parallel equivalent magnetic circuits?

Author Response

1.I have modified it.

2.This machine inherited from the Prius2010 IPM machine,whilst the V-shaped IPM rotor topology is employed as well.

3.I have modified it.

4.I have modified it

5.Due to the cross coupling effect between the two permanent magnets, the magnetic flux of the high coercivity permanent magnet will affect the magnetization state of the low coercivity permanent magnet, thus affecting the magnetic density of the air gap.
As mentioned in the article, when the two permanent magnets are connected in positive parallel, the high coercivity permanent magnet has an inhibitory effect on the low coercivity permanent magnet, which makes the magnetization state of the low coercivity permanent magnet low and produces small magnetic flux.
When the two permanent magnets are in reverse parallel, most of the magnetic flux of the high coercivity permanent magnet forms a loop with the low coercivity permanent magnet, which greatly reduces the air gap magnetic density. At the same time, the magnetic flux of the high coercivity permanent magnet will further increase the magnetization state of the low coercivity permanent magnet, so as to further reduce the air gap magnetic density.